# Novelties in Imaging of Thoracic Sarcoidosis

**DOI:** 10.3390/jcm10112222

**Published:** 2021-05-21

**Authors:** Lucio Calandriello, Rosa D’Abronzo, Giuliana Pasciuto, Giuseppe Cicchetti, Annemilia del Ciello, Alessandra Farchione, Cecilia Strappa, Riccardo Manfredi, Anna Rita Larici

**Affiliations:** 1Department of Diagnostic Imaging, Oncological Radiotherapy and Hematology, Fondazione Policlinico Universitario “A. Gemelli” IRCCS, 00168 Rome, Italy; lucio.calandriello@policlinicogemelli.it (L.C.); annemilia.delciello@policlinicogemelli.it (A.d.C.); alessandra.farchione@policlinicogemelli.it (A.F.); riccardo.manfredi@policlinicogemelli.it (R.M.); annarita.larici@policlinicogemelli.it (A.R.L.); 2Section of Radiology, Department of Radiological and Hematological Sciences, Università Cattolica del Sacro Cuore, 00168 Rome, Italy; rosadabronzo@hotmail.it (R.D.); strappacecilia@gmail.com (C.S.); 3Pulmonary Medicine Unit, Department of Medical and Surgical Sciences, Fondazione Policlinico Universitario “A. Gemelli” IRCCS, 00168 Rome, Italy; giuliana.pasciuto@policlinicogemelli.it

**Keywords:** sarcoidosis, HRCT, prognostic assessment, CXR, radiomics, chest MR

## Abstract

Sarcoidosis is a systemic granulomatous disease affecting various organs, and the lungs are the most commonly involved. According to guidelines, diagnosis relies on a consistent clinical picture, histological demonstration of non-caseating granulomas, and exclusion of other diseases with similar histological or clinical picture. Nevertheless, chest imaging plays an important role in both diagnostic assessment, allowing to avoid biopsy in some situations, and prognostic evaluation. Despite the demonstrated lower sensitivity of chest X-ray (CXR) in the evaluation of chest findings compared to high-resolution computed tomography (HRCT), CXR still retains a pivotal role in both diagnostic and prognostic assessment in sarcoidosis. Moreover, despite the huge progress made in the field of radiation dose reduction, chest magnetic resonance (MR), and quantitative imaging, very little research has focused on their application in sarcoidosis. In this review, we aim to describe the latest novelties in diagnostic and prognostic assessment of thoracic sarcoidosis and to identify the fields of research that require investigation.

## 1. Introduction

Sarcoidosis is a systemic inflammatory disease characterized by widespread development of non-caseating epithelioid cell granulomas, and the lungs are the most affected organs [1].

According to the statement on sarcoidosis released by the American Thoracic Society (ATS), the World Association for Sarcoidosis and Other Granulomatous Disorders (WASOG), and the European Respiratory Society (ERS), the diagnosis relies on a consistent clinical picture, histological demonstration of non-caseating granulomas and exclusion of other disease with similar histological or clinical picture. In this context, chest X-ray (CXR) plays a crucial role obviating the need for biopsy in patients with typical clinical presentation and typical radiographic findings, the so called Löfgren syndrome, characterized by bilateral hilar lymphadenopathies, erythema nodosum, and/or bilateral ankle arthritis [2].

Computed Tomography (CT) is more sensitive than CXR in identifying lymphadenopathies and in depicting subtle parenchymal abnormalities [3], and it is superior to CXR in identifying pulmonary fibrosis, which represents the end-stage of this disease [4]. CT is therefore extremely helpful in the diagnostic work up when there is discordance between clinical data and radiographic features or when the latter are atypical. The latest British Thoracic Society (BTS) statement on pulmonary sarcoidosis confirms the central role of CXR in diagnostic assessment and monitoring of the disease but focuses more on the utility of CT scans, that may prevent the need of invasive sampling (endobronchial ultrasound—EBUS—or transbronchial biopsies) in the presence of typical CT features. However, also typical CT findings associated with a confident clinical diagnosis might require a multidisciplinary team (MDT) discussion to assess the potential need of biopsy [4].

Prognostic assessment in sarcoidosis still relies on a four-level staging system based on radiographic appearance, conceived by Scadding in 1961, nearly sixty years ago, that goes from stage 0—normal findings—to stage IV—lung fibrosis with no chance of clinical and radiographic resolution if untreated [4,5].

Therefore, despite the advances in chest imaging modalities including the wider use of radiation-saving techniques, the ever-increasing applications of chest magnetic resonance (MR), and the introduction of objective and quantitative techniques, according to guidelines, diagnostic and prognostic assessment in sarcoidosis seems to be stuck on CXR [2,4]. In this review, we aim to describe the latest novelties in diagnostic and prognostic assessment of thoracic sarcoidosis—focusing on imaging of lung abnormalities and lymphadenopathies—and outline potential research fields, which have not been explored in this disease yet.

## 2. Diagnostic Assessment

Imaging plays a central role in both diagnosis and monitoring of thoracic sarcoidosis [4].

Typical and atypical radiographic and CT findings in sarcoidosis have been widely described in literature, with high-resolution computed tomography (HRCT) being far superior in detecting and characterizing lung disease compared to CXR [6]. Lymphadenopathies and lung parenchymal abnormalities represent the main features of thoracic sarcoidosis, followed by airway and pleural disease [3]. Cardiac and musculoskeletal abnormalities are beyond the purpose of this review; thus, they will not be discussed.

Lymphadenopathies are the most common intrathoracic manifestation of sarcoidosis (occurring in 75–80% of patients at some point in their disease) and are typically symmetrical and non-necrotic, with extremely variable degree of enlargement. Lymph node calcification is not uncommon and, when present, may be punctate, amorphous, eggshell, and popcorn-like in appearance [3]. The characteristic pattern is bilateral hilar lymphadenopathies; however, the most involved nodal stations, in descending order, are right lower paratracheal (4R), right hilar (10), subcarinal (7), subaortic (5), and interlobar (11) [3].

Typical parenchymal abnormality consists of small nodules, occurring with a perilymphatic distribution and with variable degree of involvement among individual patients, typically visible along the peribronchovascular interstitium, the pleural surfaces, including fissures, and the centrilobular regions; sarcoid granulomas frequently cause nodular thickening of the axial perihilar and peribronchovascular interstitium [6].

Nodules are usually small (diameter 1–4 mm), with irregular borders and often have a bilateral and symmetrical distribution involving mainly upper and middle lung zones. Less commonly, they may show an asymmetric involvement of the lung parenchyma or a lower lung zone predominance; exceptionally a miliary pattern may be seen [3].

Confluence or coalescence of granulomas may result in large nodules (diameter < 3 cm) or masses with ill-defined contours or regions of dense consolidation (*alveolar sarcoidosis*); occasionally they contain air bronchograms. Cavitary lesions are rare (3.4–6.8% of cases), usually indicating severe and active sarcoidosis, resulting from either ischemic necrosis or vasculitis [3,7].

A typical CT sign is the “*galaxy sign*”, characterized by a central mass surrounded by numerous small satellite nodules; less commonly the “*halo sign*” (solid lesion surrounded by a rim of ground-glass opacity) or “*reversed halo sign*” (“*atoll sign*”) may be seen [6].

Focal or patchy areas of ground-glass opacity are also possible features (18% to 83% of cases); compared to other causes of ground-glass opacity, in patients with sarcoidosis, this finding has usually an upper lobe distribution, overlaid on a background of small perilymphatic nodules, with symmetric mediastinal and hilar adenopathies [6].

Fibrotic changes indicate irreversible disease, typically involving middle and upper lung zones; imaging findings include fissure and bronchial displacement and distortion, traction bronchiectasis, distorted septal reticulation and honeycombing (which in this case usually affects upper and perihilar regions and tends to be macrocystic). The posterior displacement of the main or upper lobe bronchus with associated volume loss, particularly in posterior segment of the upper lobes, is a typical feature of fibrotic sarcoidosis [3].

Airway disease is common and may occur at any level, from the epiglottis to the peripheral bronchioles. Air trapping (defined as areas of decreased attenuation on expiratory CT) is another frequent finding related to distal airways involvement in sarcoidosis, caused by peribronchial or intraluminal granulomas obstructing small airways. Occasionally, it may be the only CT finding in sarcoid patients. Finally, pleural involvement in sarcoidosis is rare, appearing as pleural effusions, pneumothorax or pleural thickening [3].

### 2.1. Low Dose Computed Tomography (LDCT)

Despite the pivotal role of HRCT in the diagnosis of sarcoidosis, the exposure to ionizing radiation still represents an important limit of this modality, which should be taken into account especially in young patients and in those requiring repeated CT. One of the main risks related to radiation exposure is developing a radiation-induced cancer, which depends on the effective dose measured in Sieverts (Sv). Patients who undergo annual chest CT can accumulate over 100 mSv of exposure in their lifetime [8,9]. According to guidelines, surveillance CT—due to its high sensitivity in depicting lung abnormalities—is recommended in patients with unexplained deterioration in symptoms or lung function, those with “red flags” such as hemoptysis or pulmonary hypertension, or when specific conditions are suspected radiographically (such as aspergilloma) [4].

Recent advances in CT technology and implementation and diffusion of iterative reconstruction software, have led to a significant reduction of the ionizing radiation dose associated with this imaging modality and to the introduction of Low Dose Computed Tomography (LDCT)—with radiation dose ranging from 1 to 2 mSv—and UltralowDose Computed Tomography (ULDCT)—with radiation dose < 1 mSv (Table 1) [8,10,11,12,13].

LDCT, due to its greater sensitivity in detecting nodules and lung cancers compared to CXR, has been playing a central role in lung cancer screening, showing a decrease in mortality from lung cancer by about 20% compared to CXR [14,15].

LDCT is also able to detect other lung abnormalities, such as consolidations, emphysema, ground glass opacities and fibrotic changes [16,17]. Several studies have been conducted to assess the presence of ILD in smokers included in lung cancer screening population undergoing LDCT, showing promising results [17,18].

Manners et al. used ULDCT in an asbestos-exposed population to evaluate the presence of interstitial abnormalities and compared them with pulmonary function tests (PFT), demonstrating that an ultra-low-dose protocol can reliably identify interstitial changes with good inter-observer agreement and significant correlation with reduced gas transfer [10].

Nevertheless, despite the wide use of HRCT in sarcoid patients and albeit chest abnormalities in sarcoidosis—ranging from micronodules to fibrotic changes—can be virtually detected by LDCT and ULDCT, only a single case report is found in literature about the use of these techniques in sarcoid patients.

Heyer et al. performed a low-dose CT-guided transthoracic lung biopsy in a 14-year-old boy, affected by chronic pulmonary infiltrates, who unsuccessfully underwent multiple diagnostic procedures. Using a low-dose protocol with calculated total effective dose of 1.5 mSv for initial scans (scout view and a spiral scan from the diaphragms to the lung apices) and of 0.4 mSv for biopsy procedure, a diagnosis of stage III sarcoidosis was made [19].

Since the potential advantages described above and the need to keep the radiation exposure as low as possible, studies are needed to validate these radiation-saving techniques in sarcoid patients, in particular in disease monitoring and, although uncommon, in children (Figure 1).

### 2.2. Magnetic Resonance Imaging (MR)

MR—a radiation-free imaging modality—once considered inadequate in the evaluation of lung disease due to poor signal-to-noise ratio (SNR) and long scan time, has become an alternative technique in the assessment of many lung diseases in children and adults [20,21]. Latest advancements, including parallel imaging, multi-array phase coils and ultra-short echo-time techniques, have led to higher image quality and shorter scan time [8,20].

Over the last few years many studies have assessed the utility of MRI in the evaluation of lung parenchyma, particularly in interstitial lung diseases (ILDs) [20,22,23,24].

To date, the suggested standard chest MRI protocol takes approximately 15 min [23].

After localizer sequences, a T2-weighted sequence (generally on coronal plane) should be acquired using single shot techniques (i.e., half-Fourier acquisition single-shot turbo spin echo imaging, HASTE, by Siemens or single-shot fast spin echo, SS-FSE, by General Electric—GE). T2-weighted imaging is extremely helpful in the assessment of ILDs, in which inflammatory and fibrotic changes are typically interspersed; inflammatory areas, generally manifesting as acute alveolitis, are associated with high water content within the tissues and, therefore, appear as areas of high signal intensity, whereas fibrotic-predominant lesions will be hypointense [23]. T2-weighted HASTE sequence can demonstrate also other kind of pulmonary infiltrates, which will appear as hyperintense areas contrasting against the dark background of the normal lung parenchyma, including mucus and fluid accumulation and inflammatory bronchial thickening. Following T2-HASTE, a T1-weighted sequence (on axial plane) is performed using spoiled three-dimensional (3D) gradient echo sequences (GRE) (i.e., volumetric interpolated breath-hold examination, VIBE, by Siemens or fast acquisition with multiphase elliptical fast gradient echo, FAME, by GE) [23]. This sequence is particularly suited for mediastinal evaluation and to assess bulky consolidations and large areas of fibrosis in inspiration, and to detect trapped air in expiration [8,20]. Furthermore, a steady-state free-precession GRE sequence should be acquired (i.e., true fast imaging with steady-state precession, TrueFISP, by Siemens or fast imaging employing steady-state acquisition, FIESTA, by GE), which provides shorter echo and acquisition times, lower sensitivity to motion artifacts and mixed T2/T1 contrast weighting. Besides identification of pulmonary fibrosis, ground-glass opacities, and traction bronchiectasis, TrueFISP—which is “white blood” flow sensitive—demonstrates a great accuracy in detection of pulmonary embolism [23]. Finally, contrast-enhanced sequences should be included, generally T1-weighted 3D-GRE, to allow a clearer evaluation of vascular and hilar structures, pleura, and solid nodules or masses. Contrast-enhanced sequences are also extremely helpful in the evaluation of ILDs; generally, early enhancement is an indicator of active disease, whereas late-enhancement correlates with fibrotic abnormalities. This happens because inflammatory-predominant lesions have an increased extravascular interstitial volume and angiogenesis, which results in an earlier and faster enhancement compared to normal lung interstitium; late-enhancement in fibrosis, instead, might be a result of the destruction of the lung interstitial microvasculature, impairing contrast washout [23].

Chung et al. were the first group to compare HRCT to MRI in pulmonary sarcoidosis, using a multi-sequence approach based on coronal HASTE, axial fat-saturated T2-weighted fast spin-echo (T2-FSE/ BLADE), coronal TrueFISP and axial contrast-enhanced VIBE.

They established four main categories of parenchymal abnormality evaluated on both CT and MRI (parenchymal opacification, reticulation, nodules, and masses) and each of these categories was scored by lobe on a 3-point scale (0—absent, 1—less than 50% lobar involvement, and 2—greater than 50% lobar involvement). In their study, MRI and HRCT showed a good agreement, with highest correlation for parenchymal opacification (with a Spearman correlation coefficient of 0.695) and weaker correlation for small nodules (with a Spearman correlation coefficient of 0.501) owing to the inherent lower spatial resolution of MRI and, probably, the non-inclusion of STIR sequences in the MRI protocol, which have been reported to be the most sensitive in the detection of sub-centimetric pulmonary nodules. There was also poorer agreement in the lower lobes due to motion artifact [8].

Brady et al. assessed the utility of late-enhanced MRI in sarcoidosis using a specific segmented turboFLASH pulse sequence with an inversion time (TI) individually chosen to null the pulmonary arterial blood pool signal following contrast material administration. They found that late-enhanced MRI correctly identified all patients with fibrotic pulmonary sarcoidosis and, moreover, it correlated significantly with HRCT for its extent (with a Spearman correlation coefficient of 0.84) [25].

Another possible application of MRI which should be strongly encouraged is evaluation of sarcoidosis in children, where the use of ionizing radiation should be minimized. Gorkem et al. showed that there was no statistically significant difference between lung MRI-fast sequences (with average total time on the table of 10 ± 3 min) and CT in detecting thoracic findings (*p* = 0.1336, 95% Cl) [26].

Other MRI techniques have been explored in the assessment of ILDs and other subtypes of lung fibrosis, though never in sarcoid patients. These include hyperpolarized gas MRI with Xenon-129 (129Xe), a spectroscopic technique. This gas is inhaled—in pure state or mixed with oxygen or nitrogen—by the patient, who is asked to hold the breath for few seconds during acquisition. Since 129Xe has the ability to cross the alveolar interstitium into capillary blood, MRI spectroscopic techniques can evaluate diffusion limitation, by leveraging the unique chemical shift of this hyperpolarized gas in gaseous, aqueous (tissue and plasma; TP) and red blood cell (RBC) environments [27]. A reduction of RBC peak compared to TP peak was found in patients affected by idiopathic pulmonary fibrosis (IPF) suggesting a correlation with interstitial thickening [28]. This functional technique, whilst not yet explored in sarcoidosis, could provide a valuable support in the assessment and/or quantification of fibrotic changes related to sarcoidosis with possible implication for clinical management.

MRI is useful also in the assessment of mediastinal lymph nodes. It can accurately detect subtle enhancement and necrosis within mediastinal lymph nodes, which are commonly found in tuberculosis but not in sarcoidosis, helping in differential diagnosis [23].

Additionally, a characteristic MRI sign has been described for the diagnosis of sarcoid related lymph nodes. The “*dark lymph node sign*” is defined as an internal hypointense region with peripheral hyperintensity within mediastinal and/or hilar lymph nodes (relative to paravertebral muscle) on T2-FSE (BLADE) and post-gadolinium 3D-GRE (VIBE) images. This sign was present in up to 49% of patients with sarcoidosis and is probably related to areas of central nodal fibrosis [29].

### 2.3. Radiomics

The term ‘Radiomics’ refers to the process of extraction of sub-visual, quantitative image features from radiological images [30]. This field has been playing an emerging role in the assessment of several lung disease and has already shown its potential in the development of quantitative biomarkers in emphysema, ILDs and lung cancer [31]. Once selected a region of interest (ROI), it is segmented and, subsequently, analyzed in order to obtain quantitative features, which are currently divided into four classes: intensity-based, structural, texture/gradient-based, and wavelet (Figure 2).

Intensity-based features are obtained from image histogram. Histograms are graphical representations of the intensity distribution of an image, from which statistical measures are extracted (mean, median, standard deviation, kurtosis, skewness, etc.). Structural features describe basic characteristics of the ROI like shape, volume, and derivative measurements such as surface area and surface-area-to-volume ratio. Texture features and gradient features refer to spatial relationships and interactions between pixel intensities in a given local neighborhood and is usually related to heterogeneity in a specific ROI. Wavelet features enable identification of image attributes in response to different spatial frequencies [30].

One study has evaluated the diagnostic potential of radiomics in sarcoid patients, in order to differentiate tuberculous lymph nodes from sarcoid ones. These two can be distinguished by the presence of central low attenuation—indicating caseous necrosis—and a peripheral rim enhancement—indicating granulation tissue, both generally present only in tuberculosis. Nevertheless, this differential diagnosis can be challenging as central necrosis may be not always clearly visualized on CT images. In the study of Lee et al. statistically significant differences were found in the skewness and kurtosis of CT histograms between the sarcoid and tuberculous lymph nodes, as the sarcoid group showed lower negative skewness and higher positive kurtosis compared to tuberculous ones. The subgroup analysis of lymph nodes without central low attenuation also revealed statistically significant differences in other Radiomics features (namely the Feret’s diameter, perimeter, area, and circularity of lymph nodes) between the two groups [32].

## 3. Prognostic Assessment

Prognostic assessment in sarcoidosis still relies on the Scadding staging system based on CXR findings (Table 2) [4]. As showed in Table 2, this system is related to the likely disease course, with percentage of resolution in untreated patients decreasing as Scadding stage increases. Spontaneous resolution happens in 50–90% of patients with only lymph node enlargement (stage I disease). Gradual reduction in lymph node size usually occurs in the first 3 to 6 months and complete resolution may happen within 2 years. In the remaining cases, parenchymal opacities can emerge as lymph nodes decrease in size. Percentage of resolution in untreated patients decreases by nearly 20% from Scadding stage I to stage II—lymph node enlargement with parenchymal changes—and stage III—only parenchymal abnormalities. Fibrotic changes represent the end-stage of this disease (stage IV) and do not resolve [3,4] (Figure 3).

Current guidelines in management of sarcoidosis recommend serial chest radiography—evaluating possible shifts in the Scadding staging—and PFTs (pulmonary function tests) in order to detect changes in disease severity [2,4]. Nevertheless, several limitations come with this four-level staging system based on radiographic appearance, despite its wide application in the past nearly sixty years. Firstly, thoracic abnormalities may not necessarily follow the Scadding staging order [31]. Secondly, changes in the staging do not take into consideration changes in radiographic disease extent [3]. Lastly, the main limitation of the Scadding staging system is subjectivity: some features on chest radiography have variable appearance, including lung fibrosis that may range from subtle radiographic signs to end-stage diffuse fibrosis, leading to poor inter-observer agreement [3,33]. Yet, proper identification of fibrotic abnormalities is essential for an adequate management of sarcoid patients [4]. The limits of the Scadding system represent the basis for implementing new techniques and exploring new approaches to improve the prognostic evaluation of sarcoid patients just like it is happening in other ILD [34].

### From Semi-Quantitative Image Analysis to Radiomics

Over the past years, many studies have tried to overcome the aforementioned limitations of chest radiography in prognostic assessment of sarcoidosis.

First attempts were made by combining a semiquantitative radiographic scoring system—yet subjective—with clinical or physiologic data [35,36]. Zappala et al. investigated the correlation between changes in Scadding stage and extent of disease on CXR and compared them with variation in PFTs in 354 sarcoid patients, assessing forced expiratory volume in the first second (FEV1), forced vital capacity (FVC), and diffusing capacity of the lungs for carbon monoxide (DLco). They found that changes in radiographic extent of disease were more frequent than changes in Scadding stage and a poor correlation was shown between the change of radiographic stage and extension. Moreover, changes in radiographic extent of disease—and *not* in Scadding stage—significantly correlated with pulmonary function trends (*p* < 0.0005 for FEV1, FVC, DLCO) [35].

HRCT has allowed an additional improvement in prognostic assessment of sarcoid patients. Some studies have combined HRCT findings and physiological variables, which independently may predict survival, presuming that a more powerful prognostic discrimination may come from an integration of both [37,38]. Walsh et al. conceived an integrated clinicoradiological staging system for rapid risk prediction in sarcoidosis, using the CPI (Composite Physiologic Index)—a weighted index of pulmonary function variables—that correlates with extent of interstitial disease and HRCT measures of extent of fibrosis and MPAD/AAD (main pulmonary artery diameter to ascending aorta diameter ratio). CPI (threshold set at 40), MPAD/AAD (< or >1) and extent of fibrosis (threshold set at 20%) were combined to form a staging algorithm, which was found to be more predictive of mortality than any individual variable alone (HR 5·89, 2·68–10·08, *p* < 0.0001) [38].

However, these studies share a common limit, which is the underlying subjectivity in their semi-quantitative methods and poor inter-observer agreement. Over the last few years, quantitative CT (QCT), defined as the use of computerized tools to quantify disease on CT, and radiomics, with their objective and reproducible results, have been tested in the assessment of several lung disease, including lung cancer, emphysema and interstitial lung disease [30,39,40]. These tools can moreover identify and quantify imaging features that may not be recognized by human eyes [32].

Erdal et al. conceived a two-point correlation function (TPCF) based approach to assess the severity of lung involvement in sarcoid patients using CT. They developed a program to produce a lung texture score (LTS) that objectively quantified the disease extension both in terms of interstitial and alveolar processes. LTS strongly correlated with PFT (FVC, TLC, and DLCO) (*p* < 0.0001 for all comparisons), that, currently, guide the management of this disease [41]. In the study of Urbankowski et al., lung involvement in sarcoidosis was assessed using an open-source software (OsiriX Lite) to quantify CT features. They demonstrated that CT-QI (computed tomography-derived quantitative indices) differed significantly in patients with different stages of sarcoidosis, specifically value of SD_LR_ (standard deviation of lung radiodensity) was significantly higher in patients with lung fibrosis. Moreover, CT-QI correlated with FVC, FEV1, and TLC (total lung capacity) [42].

Despite the potential detection of subclinical lung disease and the excellent correlation with functional tests, the prognostic value of the proposed approaches was not assessed.

Other QCT tools have been developed, providing good results in terms of prognostic assessment and correlation with PFT, and have been tested in ILD, mainly IPF, but not in sarcoidosis. CALIPER (Computer Aided Lung Informatics for Pathology Evaluation and Rating), a QCT tool based on both 3D histogram features within a regional voxel and morphological analysis, has been tested in several fibrosing lung disease showing good agreement with PFT and prognosis. CALIPER has shown the ability to quantify the vessel-related structures (VRS), corresponding to the volume of pulmonary vessels and associated structures, such as perivascular fibrosis, which can not be quantified visually. The prognostic value of VRS has been shown in several fibrosing lung disease, including IPF, hypersensitivity pneumonitis, CTD-ILD, and unclassifiable ILD [40,43]. Even though CALIPER was not tested in sarcoidosis, it is likely that it will retain its prognostic potential also in this class of patients just like it does in the other fibrotic ILD.

Finally, only one study assessed the utility of Radiomics in sarcoidosis. Ryan et al. applied radiomic measures, obtained from HRCT, in 73 sarcoid patients and compared them with PFT. They found that global radiomic measures differed significantly between sarcoid patients and healthy controls (*p* < 0.001 for skewness, kurtosis, fractal dimension, and Geary’s C), with sarcoid ones showing lower skewness and kurtosis—likely caused by increased opacification due to parenchymal abnormalities that make distribution of Hounsfield units in HRCT more normally distributed—and lower fractal dimension, higher Moran’s I and lower Geary’s C—indicating more similar adjacent pixels likely due to nodule conglomeration and/or fibrosis. Additionally, these measures significantly correlated with PFT, with significant association between Geary’s C and FVC (*p* = 0.008) and proved to be a better predictor of functional impairment compared to the Scadding stage system [31]. Although promising as quantitative biomarkers for pulmonary sarcoidosis, it would be interesting to investigate how these results correlate with prognosis and if could overcome the inter-observer variability related to Scadding staging system. Table 3 summarizes the latest novelties in diagnostic and prognostic assessment and the potential research fields in thoracic sarcoidosis discussed in this review.

## 4. Conclusions

Imaging of sarcoidosis has been the subject of a wide number of publications over the last 60 years providing a detailed description of all the possible radiographic and CT findings associated with this disease. Nevertheless, despite the advances in thoracic imaging modalities and the introduction of new computerized tools for the analysis of chest imaging examination, diagnostic and prognostic assessment of sarcoidosis, according to guidelines, still relies on chest radiography. Some well-known techniques, including low dose CT and MRI, have not been explored enough in sarcoid patient. Other more recent techniques, including quantitative CT tools and Radiomics, are showing their potential in diagnostic and prognostic assessment of several ILD but have only marginally been tested in sarcoidosis. Future research assessing and validating these new modalities and techniques in sarcoid patients might lead to a different management of these patients allowing to obtain more information with a reduced or absent radiation exposure and to improve prognostic assessment.

## Figures and Tables

**Figure 1 jcm-10-02222-f001:**
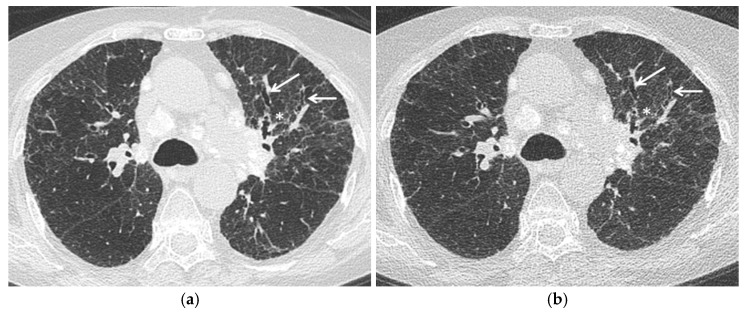
Stage IV pulmonary sarcoidosis. Comparison between (**a**) standard chest HRCT acquisition (effective dose 3.3 mSv) and (**b**) Ultra-low dose acquisition (effective dose 0.8 mSv) with a 75% radiation dose reduction. Fibrotic changes—specifically irregular interstitial thickening (asterisk) and traction bronchiectasis (arrows)—are clearly evident at Ultra-low dose acquisition. Mediastinal calcified lymph nodes are also evident.

**Figure 2 jcm-10-02222-f002:**
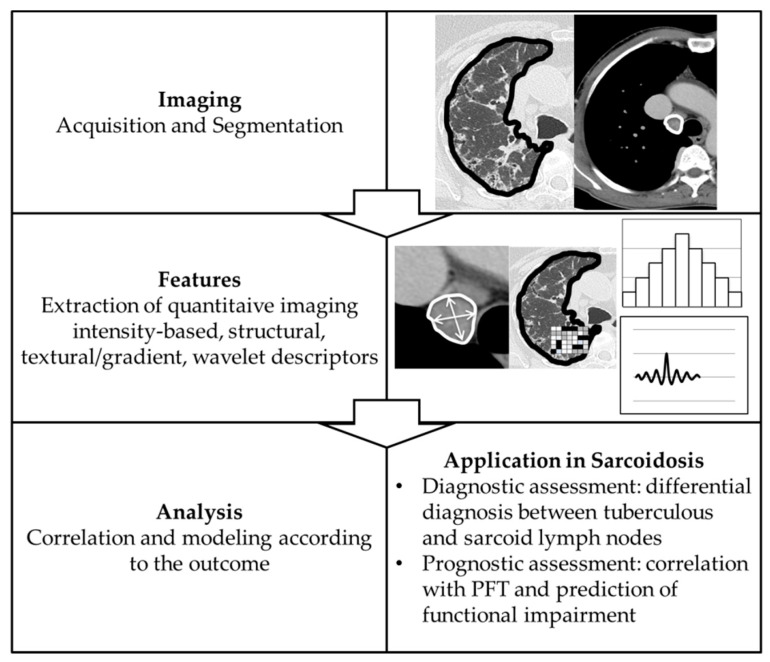
Radiomics workflow and its application in sarcoidosis.

**Figure 3 jcm-10-02222-f003:**
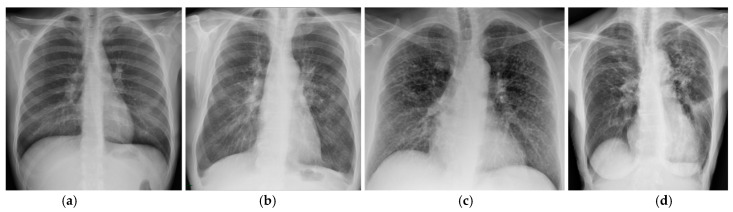
CXR of patients with thoracic sarcoidosis. (**a**) Scadding stage I showing bilateral hilar enlargement. (**b**) Scadding stage II showing bilateral hilar enlargement and parenchymal reticulation and micronodules. (**c**) Scadding stage III showing bilateral micronodules without significant hilar enlargement. (**d**) Scadding stage IV showing upward retraction of hila with bronchial distortion and reticular opacities.

**Table 1 jcm-10-02222-t001:** Effective radiation dose of chest imaging modalities (expressed in millisievert—mSv).

Image Modality	Effective Radiation Dose
MD-HRCT ^1^ (standard technique)	4–7 mSv
LDCT ^2^ (Lung Cancer Screening)	1–2 mSv
ULDCT ^3^	<1 mSv
CXR ^4^	0.05–0.24 mSv
MR ^5^	0 mSv

^1^ Multidetector high-resolution Computed Tomography; ^2^ Low Dose Computed Tomography; ^3^ Ultra-low Dose Computed Tomography; ^4^ chest X-ray; ^5^ Magnetic Resonance.

**Table 2 jcm-10-02222-t002:** Scadding staging system.

Scadding Stage	Findings (CXR ^1^)	Resolution in Untreated Patients	% of Patients at Presentation
0	Normal	-	-
I	Lymph node enlargement	50–90%	5–15%
II	I + Parenchymal changes	30–70%	45–65%
III	Parenchymal changes only	10–20%	30–40%
IV	Fibrosis	0%	5%

^1^ CXR: chest X-ray.

**Table 3 jcm-10-02222-t003:** Latest novelties in diagnostic and prognostic assessment and the potential research fields in thoracic sarcoidosis.

Technique	Novelties	Potential Research Fields
MRI	MRI and HRCT have showed a good agreement in the identification of parenchymal findings.	Hyperpolarized gas MRI with ^129^Xe is a spectroscopic technique, never tested in sarcoidosis, that can evaluate, quantify and potentially map interstitial thickening in lung fibrosis.
	Late enhancement allows to characterize fibrotic abnormalities.	
	MRI is a valid radiation free technique in children. Using fast sequences, the average total time on the table is approximately 10 min.	Despite the increasing use of MRI in evaluation and monitoring of several lung diseases, only few studies have focused on sarcoidosis. More studies are needed to validate the use in clinical practice of this radiation free technique for diagnosis and disease monitoring
	The ‘dark lymph node sign’ is a characteristic MRI sign useful for diagnosis of sarcoid lymph nodes.	
LDCT	Radiological features of thoracic sarcoidosis can be detected by LDCT. Only a single case report is present in literature about the use of LDCT in guiding transthoracic lung biopsy in a sarcoid patient.	No studies have been conducted on LDCT or ULDCT in sarcoidosis, even though the reduction of radiation exposure is an impellent issue that need to be faced, particularly for disease that are quite diffuse and require repeated exam for disease monitoring like sarcoidosis.
Quantitative CT analysis and Radiomics	Radiomic measures may be relevant in differentiating sarcoid and tuberculous lymph nodes.	No studies have investigated the correlation between quantitative analysis and radiomic measures and prognosis in sarcoidosis.
	Both quantitative CT analysis and radiomic measures can quantify disease extension and strongly correlate with pulmonary function tests.	

## Data Availability

Not applicable.

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
