# Peer review of "Novelties in Imaging of Thoracic Sarcoidosis"

_jcm, 2021, doi:10.3390/jcm10112222_

Round 1

Reviewer 1 Report

maybe you should add some figure about MRI and radionics in sarcoidosis 

Author Response

Thank you for your suggestion. Unfortunately in our institution we do not have any radiomic software working on sarcoid patients neither patients undergo routinely chest MRI for sarcoidosis. We chose to use only our own images and this is the reason for not including MRI and rdiomic figures.

Reviewer 2 Report

The article is well organized and informative.

However, there a lot of grammar errors.

e.g. the abstract:

Sarcoidosis is a systemic granulomatous disease affecting various organs, and lungs are the most commonly involved. According to guidelines, diagnosis relies on a consistent clinical picture, histological demonstration of non-caseating granulomas and exclusion of other diseases with similar histological or clinical picture. Nevertheless, chest imaging plays an important role in both diagnostic assessment, allowing to avoid biopsy in some situations, and prognostic evaluation. Despite the demonstrated lower sensitivity of chest X-ray (CXR) in the evaluation of chest findings compared to high-resolution computed tomography (HRCT), CXR still retains a pivotal role in both diagnostic and prognostic assessment in sarcoidosis. Moreover, despite the huge progresses made in the field of radiation dose reduction, chest magnetic resonance (MR) and quantitative imaging, very little research has focused on their application in sarcoidosis. In this review, we aim to describe the latest novelties in diagnostic and prognostic assessment of thoracic sarcoidosis and to identify the fields of research that require investigation .

„the lungs“

missing comma  > „clinical picture, histological demonstration of non-caseating granulomas and exclusion“

both diagnostic assessment > both diagnostic assessments

unnecessary comma> „allowing to avoid biopsy in some situations, and prognostic evaluation.

the huge progresses > the huge progress

missing comma >chest magnetic resonance (MR) and quantitative imaging

unnecessary space> investigation .

and goes on and on in the main text. It`s difficult to concentrate on the content.

Second I think the title is misleading. Your article is about imaging. It does not include pathology, lab work, and so on. And most of it is not novelties, they are the current diagnostic standard.

Author Response

Thank you very much for your suggestions. We revised the grammar in the whole article as requested. We changed the Title according to your comments. Nevertheless, at least in our institution, some techniques reported in the review, including chest MRI and radiomics, are not diagnostic standard in sarcoid patients.

Reviewer 3 Report

General comments:

The authors well reviewed and discussed the imaging modalities of thoracic sarcoidosis. Their article is likely to help readers to learn the current perspectives in the imaging modalities of thoracic sarcoidosis. They demonstrated the novelties of LDCT and MRI as imaging modalities in the field. They also demonstrated the development of radiomics in prognostic assessment of sarcoid patients, but a paucity of evidence in the prognostic assessment. The reviewer does not remark any concerns throughout the text in the present manuscript.

Author Response

Thank you very much for your comments.